# Monitoring of Thermal Aging of Aluminum Alloy via Nonlinear Propagation of Acoustic Pulses Generated and Detected by Lasers

**Mengmeng Li [1,†], Alexey M. Lomonosov [2,†] , Zhonghua Shen [1,†], Hogeon Seo [3] , Kyung-Young Jhang [4] , Vitalyi E. Gusev [5,*,†] and Chenyin Ni [6,*,†]**

1   School of Science, Nanjing University of Science and Technology, Nanjing 210094, China; mengmeng1_li@163.com (M.L.); shenzh@njust.edu.cn (Z.S.)
2   General Physics Institute, Russian Academy of Science, 119911 Moscow, Russia; lom@kapella.gpi.ru
3   Institute of Integrated Technology, Gwangju Institute of Science and Technology, Gwangju 61005, Korea; hogeony@hogeony.com
4   School of Mechanical Engineering, Hanyang University, Seoul 04763, Korea; kyjhang@hanyang.ac.kr
5   Laboratoire d'Acoustique de l'Université du Mans (LAUM), UMR-CNRS 6613, Le Mans Université, Avenue Olivier Messiaen, 72085 Le Mans, France
6   School of Electronic and Optical Engineering, Nanjing University of Science and Technology, Nanjing 210094, China
*   Correspondence: vitali.goussev@univ-lemans.fr (V.E.G.); chenyin.ni@njust.edu.cn (C.N.); Tel.: +86-025-8431-7756 (C.N.)
†   These authors contributed equally to this work.

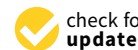

**Featured Application:** **Laser ultrasonic technique for non-destructive monitoring of material yield strength based on the evaluation of hysteretic quadratic acoustic nonlinearity of material.**

**Abstract:** Nonlinear acoustic techniques are established tools for the characterization of micro-inhomogeneous materials with higher sensitivity, compared to linear ultrasonic techniques. In particular, the evaluation of material elastic quadratic nonlinearity via the detection of the second harmonic generation by acoustic waves is known to provide an assessment of the state variation of heat treated micro-structured materials. We report on the first application for non-destructive diagnostics of material thermal aging of finite-amplitude longitudinal acoustic pulses generated and detected by lasers. Finite-amplitude longitudinal pulses were launched in aluminum alloy samples by deposited liquid-suspended carbon particles layer irradiated by a nanosecond laser source. An out-of-plane displacement at the epicenter of the opposite sample surface was measured by an interferometer. This laser ultrasonic technique provided an opportunity to study the propagation in aluminum alloys of finite-amplitude acoustic pulses with a strain up to $5 \times 10^{-3}$. The experiments revealed a signature of the hysteretic quadratic nonlinearity of micro-structured material manifested in an increase of the duration of detected acoustic pulses with an increase of their amplitude. The parameter of the hysteretic quadratic nonlinearity of the aluminum alloy (Al6061) was found to be of the order of 100 and to exhibit more than 50% variations in the process of the alloy thermal aging. By comparing the measured parameter of the hysteretic quadratic nonlinearity in aluminum alloys that were subjected to heat-treatment at 220 °C for different times (0 min, 20 min, 40 min, 1 h, 2 h, 10 h, 100 h, and 1000 h), with measurements of yield strength in same samples, it was established that the extrema in the dependence of the hysteretic nonlinearity and of the yield strength of this alloy on heat treatment time are correlated. This experimental observation provides the background for future research with the application goal of suggested nonlinear laser ultrasonic techniques for non-destructive evaluation of alloys' strength and rigidity in the process of their heat treatment.

**Keywords:** laser ultrasonics; nonlinear acoustics; hysteretic acoustic nonlinearity; thermal aging

## 1. Introduction

Thermal modifications of materials take place in many industrial fields, such as aerospace, chemical industry, nuclear power, electricity, and so on. As a consequence, it is inevitable that the components and structures served especially at elevated temperature could undergo a process of degradation of their properties and performance over time. For example, thermal aging could result in the decrease of the remaining performance life of aluminum alloys utilized in industrial structures and even threaten the security of property and personal safety. Studies have shown that the early performance degradation stage of materials occupied 80–90% of their fatigue life [1–3]. Therefore, following increasing concerns about safety, the remaining life time and structural health of the materials should be non-destructively monitored with a highest possible sensitivity.

Nondestructive evaluation techniques, for instance, X-ray diffraction, acoustic emission, ultrasonic attenuation, and velocity measurement, etc., are implemented for the detection and the characterization of thermal aging and damage in metal materials. In recent years, linear laser ultrasonic techniques have been widely used in the non-destructive testing of physical and mechanical properties of materials [4–13]. However, acoustic parameter variations measured by linear ultrasonic technology are relatively small in response to early damage and change of materials and structures [14–17]. Nonlinear acoustic techniques are widely accepted for their higher sensitivity to the state of micro-structured and damaged materials, compared with linear acoustic techniques [14,17–22]. In particular, recent studies had theoretically and experimentally confirmed that nonlinear acoustics was an effective and sensitive tool of nondestructive characterization for various thermally treated materials [17,23–34]. In comparison with other wave modes used in nonlinear acoustics to detect thermal modification of material state, for example Rayleigh waves [32,35–39] and Lamb waves [40–43], bulk waves [28,44–48] were extensively utilized. Although different types of contacting transducers are commonly used to generate and detect acoustic waves in the samples, generation and detection of acoustic waves by lasers is now applied more frequently in non-destructive detection of mechanical and physical properties of the materials. It was demonstrated that interaction between acoustic waves and materials with microstructural changes could be monitored optically [49–52]. An in-contact but nondestructive method for laser generation of high-amplitude acoustic pulses was developed [53–55]. In this LSCP (liquid-suspended carbon particles) based method, bulk finite-amplitude longitudinal acoustic waves were generated by the absorption of nanosecond laser pulses in a 50 μm thick water suspension of carbon particles, deposited on the sample surface.

In this paper, we describe the first application of all-optical generation and detection of bulk finite-amplitude longitudinal acoustic waves for measurements of acoustic nonlinearities of aluminum alloys of different thermal aging times. Finite-amplitude acoustic waves were generated on the front surface of aluminum alloy plates by applying the LSCP based method. An optical interferometer was used to detect the out-of-plane displacement of the rear surface of the plate caused by the arrival of laser-generated acoustic pulses. Aluminum alloy plates (Al6061) were heat-treated with different time durations of 0 min, 20 min, 40 min, 1 h, 2 h, 10 h, 100 h, and 1000 h [56]. It was observed that in, all evaluated samples, the duration of detected strain pulses increased with the increase of their amplitude, which is a known manifestation of the hysteretic nonlinearity of the material [57,58]. Measurements of the hysteretic quadratic nonlinearity were compared, by us, with the characterization of the elastic quadratic nonlinearity of the same samples conducted earlier [56] via monitoring of the second harmonic generation in sinusoidal surface acoustic waves and correlated with measurements of yield strength [56]. The revealed correlation of the extrema in the dependence of the hysteretic nonlinearity and of the yield strength of this aluminum alloy on the heat treatment time provides the foundation for future research, with the goal of applying the developed nonlinear laser ultrasonic

technique for non-destructive evaluation of alloys' strength and rigidity variations in the process of their heat treatment or thermal damage.

## 2. Materials and Methods

### 2.1. Mechanism of Finite-Amplitude Acoustic Wave Generation

When a laser irradiates the sample surface, a portion of laser energy is absorbed by the sample and converted into heat, causing thermal expansion of the photo-excited region. Accordingly, acoustic waves, such as Rayleigh waves, bulk longitudinal waves, and bulk shear waves, can be generated. To generate finite-amplitude acoustic waves, a thin LSCP layer was deposited on the sample surface and covered with a transparent glass plate. A schematic illustration of the sample with the liquid layer irradiated by the laser is presented in Figure 1. When the pump laser irradiates the LSCP layer, the laser energy absorption in the layer induces significant heating. The subsequent thermal expansion generates a strong transient pressure applied normally to the sample surface. The amplitude of the longitudinal acoustic pulse launched by the LSCP layer in the material could be non-destructively increased by 20–50 times relative to that generated, even with the destructive ablative mechanism. Importantly, almost complete absorption of the laser energy by the LSCP layer prevents the sample surface from laser damage. Bulk longitudinal waves of the maximal amplitude launched by the surface-applied pressure is in the direction normal to the sample surface. Thus, the out-of-plane displacement can be easily detected at the opposite surface of the sample by the interferometer.

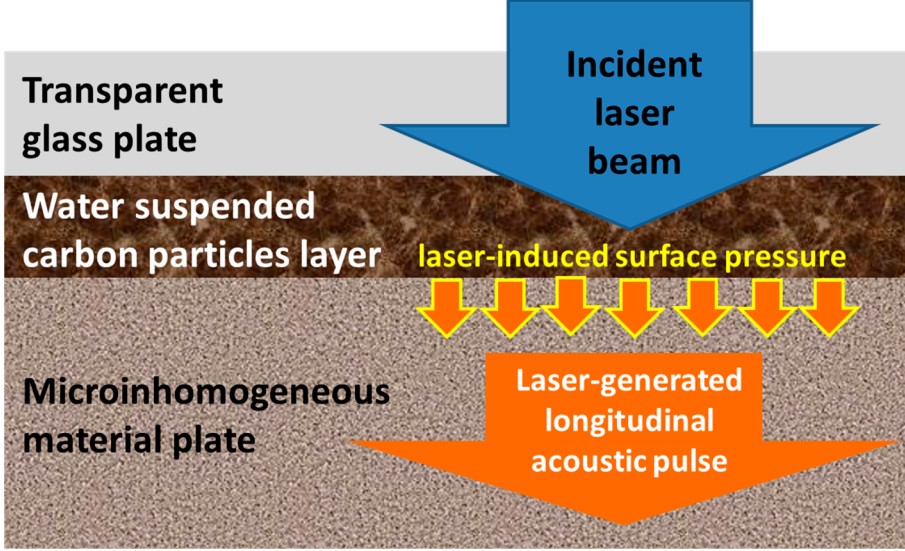

**Figure 1.** Diagrammatic sketch of the processes taking place when laser irradiates the liquid layer constrained by a transparent solid.

### 2.2. The Experimental Setup and Procedures

The samples used in experiments are 6000-series Al-Mg-Si alloy plates with the dimension of 38 mm × 20 mm × 3 mm. They were cut from an aluminum alloy block (Al6061-T6). The roughness and flatness of samples were standardized by keeping the same machining conditions. The heat-treatment processes applied to the aluminum plates was presented schematically in Figure 1 of Reference [56]. Before inducing thermal aging to the specimens, they were solution heat-treated for 4 h at 540 °C and then water-cooled quickly to put preexisting precipitating microscopic particles back into the solution, trapping the precipitates in the solution and yielding a quasi-homogeneous material state. Each sample was then heat-treated at 220 °C, for different heat-treatment times of 0 min, 20 min, 40 min, 1 h, 2 h, 10 h, 100 h and 1000 h. As a result, these samples were prepared with gradually increasing levels of thermal aging.

For experimental evaluation of the acoustic nonlinearity of thermally aged aluminum alloys, it is essential to nondestructively generate and detect finite-amplitude acoustic waves. The experimental configuration is depicted schematically in Figure 2. During the experimental process, plate-type samples were oriented horizontally and were mounted on a translation stage to adjust the laser generating position in the x-y plane. The transient stresses in the thin LSCP layer was generated by using a pulsed laser (max output power of 4 J) at the wavelength of 1064 nm, with pulse duration of 10 ns. A combination of a half wave plate and a polarized beam splitter (PBS) was implemented to tune the output laser energy. A transparent glass plate placed before the convex lens was used to reflect a portion of laser energy to the energy meter. The laser energy irradiating directly on the top surface of the tested structure was monitored in real-time through the measured ratio between the energy transmitted to the sample surface and that reflected to the energy meter. The diameter of the generation laser spot was ~3.0 mm. The out-of-plane displacement at the bottom surface was detected by an OFV-5000 interferometer (Polytec, Waldbronn, Germany, 2007), produced by Polytec, operating at 633 nm optical wavelength. The frequency range where the interferometer can be used varies from near DC to 20 MHz. A 1064 nm filter is placed in front of the interferometer to eliminate the influence of the generation laser. The detection laser was focused on the bottom surface to a diameter of approximately 80 μm to 100 μm. Finally, detected signals were displayed on a RIGOL DS4024 digital storage oscilloscope.

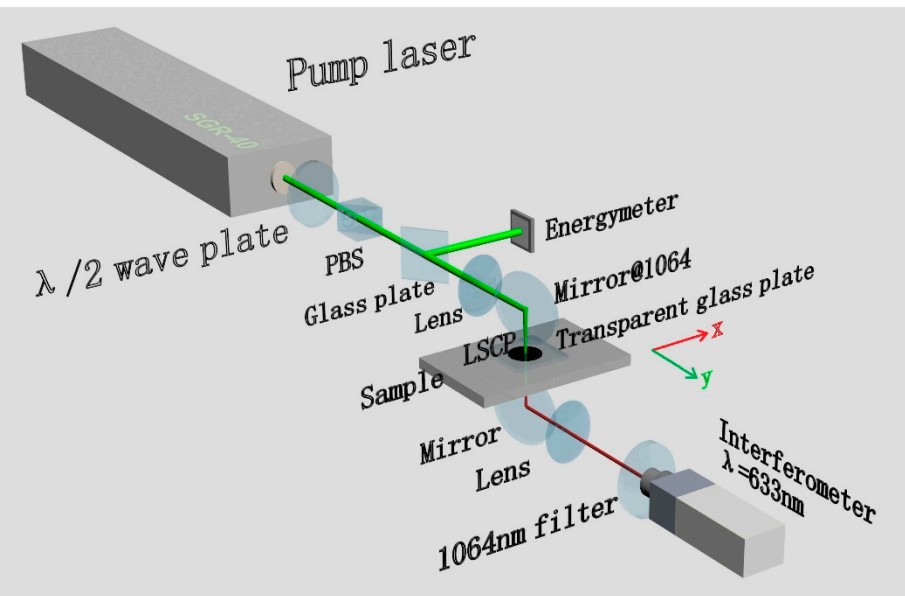

**Figure 2.** The scheme of the experimental setup.

## 3. Results

### 3.1. The Generation of Finite-Amplitude Acoustic Wave with Liquid-Suspended Carbon Particles (LSCP) Layer

As illustrated in Figure 3, two measurements of out-of-plane displacements, represented by black color and red color, were obtained at the epicenter of the rear sample surface by the interferometer. The black curve is the result without the application of the LSCP layer at a laser power density of 61.8 MW/cm$^2$ and the red curve is the result with the LSCP layer and the glass cover plate at a laser power density of 18.1 MW/cm$^2$. Based on the knowledge of the longitudinal wave velocity in these samples [56] and the thickness of the sample, the equidistant peaks marked by red arrows on the displacement curves in Figure 3 can be identified with the arrivals in the epicenter on the rear surface of the sample of the wave, propagating directly from the front surface of the sample and its subsequent four echoes. As a comparison, the amplitude of the direct longitudinal wave generated with the

use of the LSPC layer is approximately nine times larger than that generated without its application (Figure 3), while laser fluence in the first case was just one fourth of that in the second case. Thus, with the optoacoustic transducer based on the confined LSCP layer, the amplitude of the acoustic waves can be increased by more than an order of magnitude.

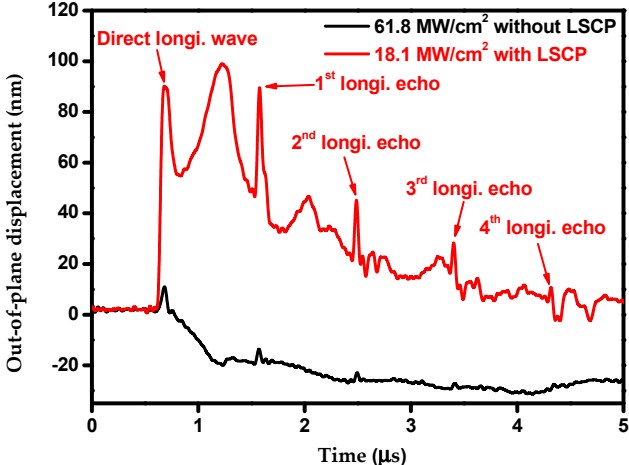

**Figure 3.** Signals with and without application of the liquid-suspended carbon particles (LSCP) layer for optoacoustic conversion at the same generation point.

### 3.2. Distortion of Direct Longitudinal Waves

The experiments demonstrated that, in all tested samples undergoing different heat-treatment times, profiles of detected acoustic pulses exhibited systematic modifications with increasing laser power densities applied for the photo-excitation of acoustic waves. As an example, we present in Figure 4 detected profiles of surface displacement and evaluated profiles of particle velocity in the direct longitudinal acoustic pulse detected in the sample subjected to thermal aging for 2 h.

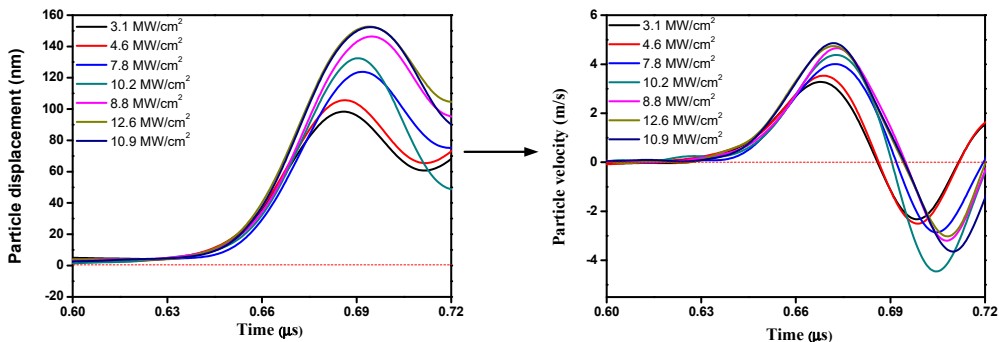

**Figure 4.** The transformation of profiles of the particle displacement (**left**) and of the particle velocity (**right**) in longitudinal acoustic pulses arriving to the rear (detection) surface of the plate directly from the front (generation) surface of the plate with increasing laser power densities applied for the excitation of the LSCP layer.

Profiles of the particle velocity presented in the right part of Figure 4 clearly reveal the following tendencies in the acoustic pulse evolution with increasing laser power: (a) There is an accumulating delay of the pulse peak arrival and (b) there is an accumulating delay in the end time of the first positive (compressional) phase of the particle velocity profile. Both these tendencies are known manifestations of the influence of the hysteretic nonlinearity of the materials on the propagation of the acoustic pulses, as theoretically predicted for longitudinal waves in micro-inhomogeneous media [57] and convincingly experimentally observed for torsional pulses in the chains of mechanical

beads [58]. Theories of longitudinal acoustic pulse propagation in hysteretic media [57,59] have been earlier applied for the explanation of the broadening, without the formation of any shock fronts, of longitudinal acoustic pulses launched in annealed aluminum by an electromagnetic-induction type transducer in comparison with those launched in non-annealed aluminum [60]. The explanation of the nonlinear transformation of the profile of unipolar acoustic pulses in aluminum as a function of the annealing temperature, applied during the preparation of the specimens, for a fixed distance of the propagation was based on the assumption that the parameter of hysteretic quadratic nonlinearity, *h*, increases with the rise of the annealing temperature. Note that this assumption is in accordance with established dependence on annealing process characteristics of the parameter *h* in copper [21,61]. The theory in [57] has also provided insight in the observed nonlinear transformation of laser-generated longitudinal acoustic pulses in cracked rocks [62].

### 3.3. Quantitative Evaluation of the Hysteretic Quadratic Nonlinearity of Aluminum Alloy

Theoretical predictions and experimental observations for acoustic pulse propagation in the media with hysteretic quadratic nonlinearity indicate that the end-time of the first (compressional) phase of the particle velocity profile in an acoustic pulse exhibits more pronounced variations as a function of both pulse amplitude and propagation distance [57,58] than the time of the particle velocity peak arrival. This fact could be appreciated by the examination of the acoustic pulse profile transformation depicted in Figures 5, 7 and 8 in Reference [57] and Figure 4 in Reference [58]. That is why, for the determination of the hysteretic parameter of specimens, we measured variations in the end-time of the detected leading phase of the particle velocity profile as a function of detected peak velocity amplitude. The results of these measurements for all samples subjected to different heat treatment durations are presented in Figure 5. Note that not only the differences in hysteretic acoustic nonlinearity of differently aged samples but also the differences in sample thicknesses contribute to the variations of the arrival times in Figure 5. However, independently of the sample's thickness, the results in Figure 5 explicitly demonstrate the tendency that in all characterized samples the growth of the amplitude of acoustic pulses results in a delayed end-time of the leading phase.

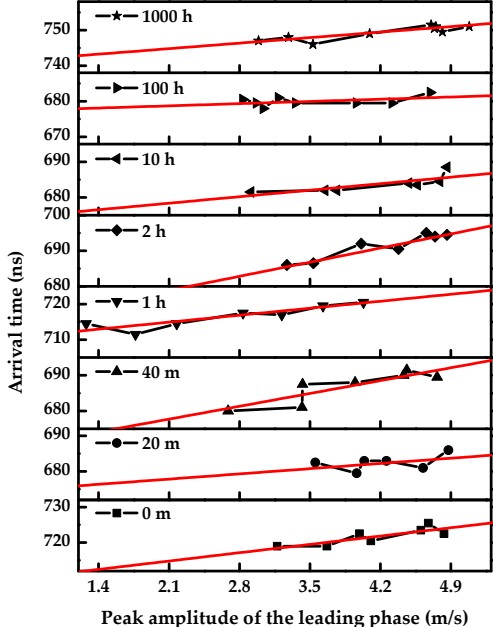

**Figure 5.** Variations in the end-time of the leading phase of the particle velocity profile in detected acoustic pulses as a function of the detected amplitude of this phase. Straight red lines provide the best linear fits for experimental points.

In theory [57] the local shift in end-time of the leading phase, $d\tau_0$, is proportional to the parameter of the hysteretic quadratic nonlinearity, $h$, to the local amplitude of the leading phase of propagating acoustic pulse, $V(x)$, and to the increase in propagation distance of the acoustic pulse, $dx$ (see, as an example, Equation (31) in Reference [57]), as follows: $d\tau_0(x) = \left[h/\left(2c_L^2\right)\right]V(x)dx$. Here $c_L$ denotes linear longitudinal acoustic velocity in the medium. In the case of weak manifestations of the hysteretic quadratic nonlinearity, as it is observed in Figure 5, the integration of the above relation along the propagation distance of an acoustic pulse leads in the first leading approximation to the following: $\tau_0(H) = \left[h/\left(2c_L^2\right)\right]V(H)H$, where $H$ denotes the thickness of the aluminum alloy plate in our experiments. Consequently, the parameter of the hysteretic quadratic nonlinearity of the material can be found in our experiments by evaluating the slope, in linear dependence, of the measured end-time of the leading phase on the measured amplitude, $V_0(H)$, of the following phase:

$$h = (4c_L^2/H)[\tau_0(H)/V_0(H)] \tag{1}$$

In deriving Equation (1) we took into account that the particle velocity at the mechanically free surface of our samples is twice larger than in propagating acoustic wave, $V(H) = V_0(H)/2$. To evaluate the hysteretic parameter in the frame of Equation (1) we used sound velocities documented for our samples in Reference [56], our measurements of the aluminum alloy plate thickness, and the slopes, $\tau_0(H)/V_0(H)$, of the best linear fit to experimental data presented in Figure 5. The evaluated dependence of the hysteretic quadratic nonlinearity on the aging time of the aluminum alloy is presented in Figure 6.

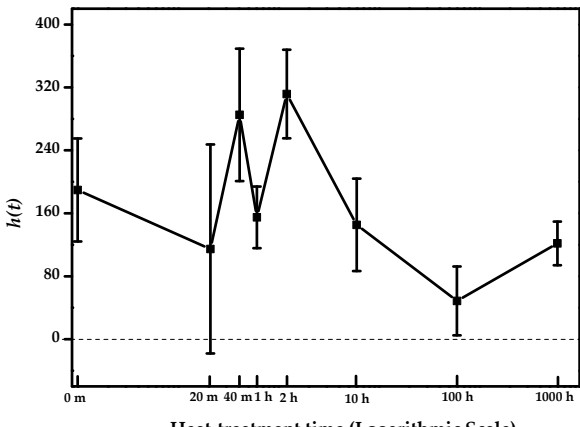

**Figure 6.** The dependence of the parameter of hysteretic quadratic nonlinearity on the time of thermal aging of aluminum alloy at 220 °C. Vertical bars provide a 70% confidence interval for red straight-line fitting in Figure 5.

Experimental results presented in Figure 6 indicate positive values of the hysteretic parameter, $h$, as it is expected theoretically [63–65] for all possible materials. They reveal values of the parameter of hysteretic quadratic nonlinearity, which are at least an order of magnitude larger than the values of the parameter of elastic quadratic nonlinearity in homogeneous solids [66]. The variations of the parameter $h$ of more than 50%, as a function of aging time (Figure 6), demonstrate its high sensitivity to the state of the thermally treated alloy. It is worth noting here that variations in the same samples of linear velocity of longitudinal acoustic waves, as a function of aging time, do not exceed 1% (Table 2 in Ref. [56]).

## 4. Discussion

It is well established theoretically and supported experimentally that typical integer-power-law elastic nonlinearities, i.e., for example quadratic and cubic, cannot induce the temporal shifts

of zero points in the acoustic wave velocity profile unless a weak shock front is formed [67,68]. The experimentally observed shift in the end-point of the leading phase of a longitudinal acoustic pulse is a fingerprint of the hysteretic quadratic nonlinearity, similar to others, such as the linear shift with the vibration amplitude of the resonance frequency of micro-inhomogeneous bars [63,69] and the generation of the third harmonic with the amplitude proportional to the square of fundamental wave amplitude in some experiments [61].

First, we evaluated the characteristic length scale for a substantial influence of the hysteretic quadratic nonlinearity on the propagation of longitudinal acoustic pulses in our experiments. The so-called nonlinear length, $x_{NL}$, is defined [57,58,65] for the hysteretic quadratic nonlinearity by the same combination of material parameters, as in the case of the elastic quadratic nonlinearity [65,67], with a replacement of the parameter of the elastic nonlinearity by the parameter of the hysteretic nonlinearity, as follows:

$$x_{NL} = 2c_L^2\tau/(hV) \tag{2}$$

where $\tau$ and $V$ are the characteristic duration of acoustic transient and the characteristic particle velocity, respectively. Substituting in Equation (2) a half of the maximal particle velocity ($V \approx 2.5$ m/s) from Figure 5, the characteristic duration of the leading phase ($\tau \approx 55$ ns) from Figure 5, the maximal measured hysteretic parameter ($h \approx 300$) from Figure 6 and the acoustic wave velocity $c_L \approx 6,400$ m/s from Reference [56], we estimate $x_{NL} \approx 6\ mm \approx 2H$ in our experiments. Thus, Equation (1) can be considered a good approximation for the evaluation of $h$ and our theoretical approach for the estimation of $h$ from experimental data is self-consistent.

Second, in Figure 7, we compare the dependencies on the duration of aluminum alloy thermal aging of the parameter of the hysteretic quadratic nonlinearity and of the yield strength. The latter was measured for the same samples in Reference [56]. The comparison reveals that aging times for two local maxima of the hysteretic parameter correlate with the position of the minima and the maxima in the variations of yield strength as a function of the thermal aging time of the aluminum alloy (Al6061).

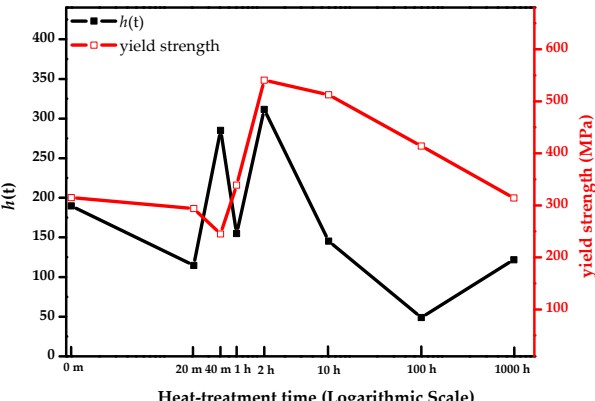

**Figure 7.** Dependence on the heat-treatment time of the measured parameter of hysteretic quadratic nonlinearity, *h*, and of the yield strength (adopted from Reference [56]).

A similar correlation was revealed earlier between the maxima of the elastic quadratic nonlinearity in longitudinal waves and the Vickers hardness of another aluminum alloy, Al2024, in Reference [70]. In both cases one of the nonlinearity maxima coincides with the maximum of the yield strength (hardness), while the preceding local minimum in the nonlinearity is achieved at an aging time located between the times of minimum and maximum yield strength/hardness (compare Figure 7 above and Figure 1 in Reference [70]). The fact of the correlation between the local maximum in the parameter of elastic quadratic nonlinear with the absolute maximum in the Vickers hardness in their dependences on thermal aging time was also documented for the steel alloy [23]. There exist theoretical developments [29,70], based on the analysis of alloy precipitation phases and the effect

of precipitate coherency strains on the acoustic harmonic generation, which provide insight in experimentally observed variations of the quadratic elastic nonlinearity with precipitate heat treatment time. Our experimental observations call for the development of the theory explaining microscopic reasons for, documented by us, strong variations in the hysteretic quadratic nonlinearity of the alloys subjected to thermal aging and their correlations with the variations of the yield strength.

Third, in Figure 8, we compare relative acoustic nonlinearity parameters between intact and aged states of the hysteretic quadratic nonlinearity, $\overline{h} \equiv [h(t) - h(t = 0)]/h(t = 0)$ for longitudinal acoustic waves, evaluated by us, using the data presented in Figure 6, and of the elastic quadratic nonlinearity for Rayleigh surface acoustic waves, $\overline{\beta} \equiv (\beta_{\text{aged}} - \beta_{\text{intact}})/\beta_{\text{intact}}$, measured in the same samples in Reference [56]. In Reference [56] the measurements were conducted in two different experimental configurations. In the first configuration, denoted as PZT-PZT in Figure 8 both the generation of the fundamental tone and the detection of the second harmonic were accomplished by piezoelectric wedge transducers. In the second configuration, denoted as LASER-PZT in Figure 8, the generation of the surface acoustic wave packet was achieved by a pulsed layer through a line-arrayed slit mask.

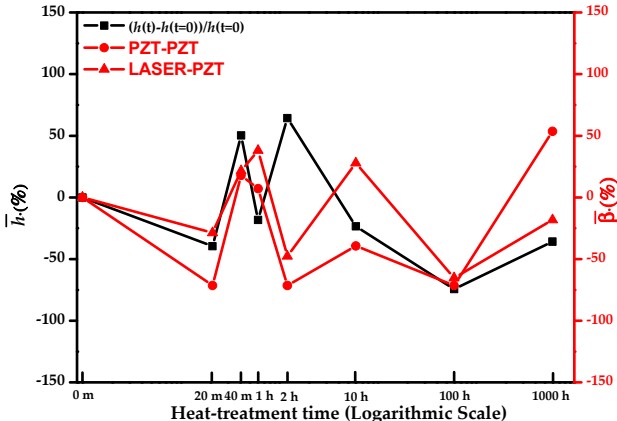

**Figure 8.** Relative variations of parameter of hysteretic nonlinearity for longitudinal acoustic waves, $\overline{h} \equiv [h(t) - h(t = 0)]/h(t = 0)$, and of parameter of elastic quadratic nonlinearity for the Rayleigh-type surface acoustic wave, $\overline{\beta} \equiv \left(\beta_{\text{aged}} - \beta_{\text{intact}}\right)/\beta_{\text{intact}}$ (adopted from Reference [56]).

The comparison reveals that in the aluminum alloy (Al6061) the parameter of the hysteretic quadratic nonlinearity for longitudinal waves exhibits relative variations, which are of the same order of magnitude as relative variations of parameter of the elastic quadratic nonlinearity for Rayleigh-type surface acoustic waves [56]. Figure 8 demonstrates that relative changes of the hysteretic and of the elastic quadratic nonlinearities are in phase for thermal aging durations $t \leq 40$ min and $t \geq 10$ h. The most interesting observations are the anti-phase variations of these two nonlinear parameters at $t = 2$ h, the time which in accordance to Figure 7 corresponds to the maximum of the yield strength in the thermally treated aluminum alloy (Al6061). Unfortunately, similar to the case of the hysteretic quadratic nonlinearity for longitudinal acoustic waves, to our knowledge, there are currently no theories of the elastic quadratic nonlinearity for surface acoustic waves in thermally treated metal alloys. However, we hope that the accumulation of experimental data on both types of the nonlinearities in the same thermally aged alloy would make the challenging theoretical developments more requested and valuable. In its current state, the proposed laser-based nonlinear acoustic method is obviously unable to predict the aging in a material that there is no previous knowledge about. For this advanced application important theoretical investigations should be undertaken to establish a link between the modifications of hysteretic quadratic nonlinearity and thermal load applied to the material. However, it could now already be possible to take advantage of the high sensitivity of the proposed technique to thermal aging by calibrating nonlinear acoustic responses for particular materials.

## 5. Conclusions

The measurements of the parameter of hysteretic quadratic nonlinearity were accomplished in aluminum alloy specimens subjected to different increasing durations of thermal aging at elevated temperatures for the first time. Correlations revealed in the variations of the hysteretic nonlinearity of the material and of the yield strength of the material in the process of its thermal aging provide the potential for future research, with the goal of applying developed laser ultrasonic techniques for non-destructive monitoring of material strength when it is modified in thermally-induced processes (aging, fatigue, damage). These application perspectives are similar to those revealed earlier for elastic quadratic nonlinearity of aluminum alloys as an indicator of microstructural changes in the material [56,70]. Currently, the progress in the application of nonlinear acoustic techniques based on bulk and surface acoustic waves is impeded by the absence of theories relating to the variations of the elastic nonlinearity parameter for surface acoustic waves and of the hysteretic nonlinearity parameter for bulk longitudinal acoustic waves with microstructural changes taking place in the sequence of precipitation phases [71–75], induced by the thermal treatment of the alloy.

**Author Contributions:** Conceptualization, C.N., V.E.G., and K.-Y.J.; methodology, V.E.G. and C.N.; software, M.L. and C.N.; validation, M.L., A.M.L., and C.N.; formal analysis, M.L. and V.E.G.; investigation, M.L., C.N., and A.M.L.; resources, K.-Y.J., H.S., Z.S., and C.N.; data curation, A.M.L., C.N., and M.L.; writing—original draft preparation, C.N., V.E.G., and M.L.; writing—review and editing, V.E.G., C.N., M.L., Z.S., K.-Y.J., and H.S.; visualization, M.L. and C.N.; supervision, Z.S., V.E.G., and K.-Y.J.; project administration, C.N., Z.S., and K.-Y.J.; funding acquisition, Z.S., C.N., and K.-Y.J.

**Funding:** This research was funded by the National Natural Science Foundation of China (Grant No. 61627802), National Natural Science Foundation of China (Grant No. 61405093), Natural Science Foundation of Jiangsu Province (Grant No. BK20140771), and the Fundamental Research Funds for the Central Universities (Grant No. 30916014112-001).

**Acknowledgments:** The authors would like to thank Zhihong Xu of Nanjing University of Science and Technology for providing the interferometer.

**Conflicts of Interest:** The authors declare no conflict of interest.

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
