# Peer review of "Monitoring of Thermal Aging of Aluminum Alloy via Nonlinear Propagation of Acoustic Pulses Generated and Detected by Lasers"

_applsci, doi:10.3390/app9061191_

Round 1
Reviewer 1 Report
Thank you for the discussion of my queries.
Author Response
Reply to the Reviewer 1
We are glad to hear that the Reviewer 1 has been satisfied by our revision of the manuscript.
With best regards
The authors

Reviewer 2 Report
Overall presentation of the problem and method is appropriate. However, the "so what" question must be answered more clearly. You detected the aging in material based on nonlinear UT, then how can you predict am unknown aging in a material that you don't have previous knowledge about? How can you distinguished between aging and other structural changes in the material?
Some other writing comments:
Introduction-Page2-Line69: Change "more and more frequently" to "more frequently"
Introduction-Page2-Line73: Change "(for liquid-suspended...)" to"(liquid-suspended ...)"
Figure 5-Page 6: Vertical scale for graphs are not consistent and it is hard to draw a conclusion
Author Response
Reply to the Reviewer 2
Point 1: Overall presentation of the problem and method is appropriate. However, the "so what" question must be answered more clearly. You detected the aging in material based on nonlinear UT, then how can you predict am unknown aging in a material that you don't have previous knowledge about? How can you distinguish between aging and other structural changes in the material?
Response 1: We really appreciate that the Reviewer 2 has accepted that our “overall presentation of the problem and method is appropriate”.
Following the suggestion of the Reviewer we have included in the Discussion/Conclusion in page 10 line 312 the following statement:
“In its current state the proposed laser-based nonlinear acoustic method is obviously unable to predict the aging in a material that there is no previous knowledge about. For this advanced application important theoretical investigations should be undertaken to establish a link between the modifications of hysteretic quadratic nonlinearity and thermal load applied to the material. However already now it could be possible to take advantage of high sensitivity of the proposed technique to thermal aging by calibrating nonlinear acoustic response for particular materials.”
Point 2: Some other writing comments:
Introduction-Page2-Line69: Change "more and more frequently" to "more frequently"
Introduction-Page2-Line73: Change "(for liquid-suspended...)" to"(liquid-suspended ...)"
Figure 5-Page 6: Vertical scale for graphs are not consistent and it is hard to draw a conclusion
Response 2:
We are grateful to the Reviewer for the suggested English improvements. We have introduced the requested corrections.
The vertical scales for the graphs in Figure 5 are the same. The different arrival times in the graphs correspond to the experimentally measured arrival times. They are different not only due to hysteretic acoustic nonlinearity but also because the samples have different thicknesses. We have added in page 6 line 203 the comment:
“Note that not only the differences in hysteretic acoustic nonlinearity of differently aged samples but also the differences in sample thicknesses are contributing to the variations of the arrival times in Fig. 5.”
However, the important information on nonlinear acoustic phenomenon is contained not in the arrival times but in the slopes of the fitting straight lines in Fig. 5 and is practically independent of the sample thickness for the samples of nearly equal thickness (as it is in our experiments). This follows from Eq. (1) and the fact that in our experiments the thickness of the samples varies by less than 5%. So, in our opinion, there is no need to hide/delete the variations of the sample thickness in Fig.5, because this could be misleading for the readers.
We hope that the Reviewer 2 would be satisfied by a minor revision of the manuscript that has been undertaken by us following his comments.
With best regards
The authors

This manuscript is a resubmission of an earlier submission. The following is a list of the peer review reports and author responses from that submission.
Round 1
Reviewer 1 Report
I found this to be a very interesting and well-written study.
I feel that the links between the h(t) peaks and the yield strength could well be co-incidental (the confidence bars are long) but the paper does not over-sell the link. It is a point that should definitely be discussed.
I am happy to recommend publication.
Reviewer 2 Report
This manuscript describes a nonlinear acoustic technique for the characterization of thermal aging in aluminum alloy Al6061. The authors claim that the proposed technique offers higher sensitivity than achievable by linear ultrasonic techniques, although no explicit evidence is actually presented to show that the measured nonlinear parameter is more sensitive to thermal aging than easily measurable linear parameters, such as velocity or attenuation. I found the paper interesting from a poorly acoustic point of view, but inappropriate for publication in MDPI’s Applied Sciences. A new nondestructive evaluation (NDE) technique based on noncontact optical measurement would certainly be of interest to the readers of the journal except for two major problems discussed separately below.
First, far simpler and far more accurate eddy current conductivity measurements have been routinely used for monitoring thermal aging of aluminum alloys for a long time and numerous studies found that the electrical conductivity and mechanical properties like hardness and yield strength respond in a complementary manner to the varying microstructures that form during different stages of the aging process (see, e.g., M. Rosen, et al. Mat. Sci. Engg. 53, 191 (1982) and numerous similar studies since then). So even if we disregard the various practical difficulties of conducting truly nondestructive thermal aging measurements using the proposed technique, it doesn’t really matter whether it performs better than linear ultrasonic techniques if the same NDE problem is much more suitable for routine eddy current inspection.
Second, the proposed technique exhibits rather modest sensitivity to thermal aging. A quick look at Figure 7, especially if one also considers the humongous error bars shown only in Figure 6, suggests that the correlation between the hysteretic quadratic nonlinearity parameter (h) and the heat treatment time and/or yield strength is highly questionable. Although the authors failed to provide a quantitative statistical analysis of these correlations, one might just say that the total change in h is roughly the same as its uncertainly level, i.e., its coefficient of determination is negligible and certainly insufficient for the targeted purpose of nondestructive materials characterization.
I should also mention that the manuscript is written in very poor English. I strongly recommend that next time the authors have their manuscript language edited by one of the numerous commercially available professional agencies offering such services. At several places technical terms are misspelled so that the text is barely intelligible. For example,
- on page 2, “depravation of their properties” probably should be “degradation of their properties”
- also on page 2, “stain pulses” probably should be “strain pulses”
- on page 7, I couldn’t really decipher what “the propagation distance of the acoustic leads in the first leading approximation to” refers to
- also on page 7, “we used the sound velocities documents for our samples in Reference [56]” probably should be “we used the sound velocities listed in [56] for our samples”
- on page “integer-power-low” should be “integer-power-law”
In summary, I cannot recommend this manuscript for publication in Applied Sciences.